# Consensus on the descriptors, definitions, and reporting methods for heading in football studies: A Delphi study

Kerry Peek[1,2]*, Andrew G. Ross[1,3], Paula R. Williamson[4], Julia Georgieva[1,5], Thor Einar Andersen[6,7], Tim Meyer[8], Vincent Gouttebarge[9,10,11], Sara Dahlen[6], Mike Clarke[12], Andreas Serner[2]

**1** Sydney School of Health Sciences, Faculty of Medicine and Health, The University of Sydney, Sydney, Australia, **2** FIFA Medical, Fédération Internationale de Football Association, Zurich, Switzerland, **3** Physiotherapy Department, College of Sport, Health and Engineering, Victoria University, Melbourne, Australia, **4** Department of Health Data Science, University of Liverpool, Liverpool, United Kingdom, **5** Curtin School of Allied Health, Curtin University, Perth, Australia, **6** Oslo Sports Trauma Research Center, the Norwegian School of Sport Sciences, Oslo, Norway, **7** The Norwegian Football Association's Sports Medical Centre, Oslo, Norway, **8** Institute of Sports and Preventive Medicine, Saarland University, Saarbrucken, Germany, **9** Department of Orthopedic Surgery and Sports Medicine, Amsterdam UMC location University of Amsterdam, Amsterdam, The Netherlands, **10** Section Sports Medicine, Faculty of Health Sciences, University of Pretoria, Pretoria, South Africa, **11** Football Players Worldwide (FIFPRO), Hoofddorp, The Netherlands, **12** Centre for Public Health, Queen's University Belfast, Belfast, United Kingdom

* Kerry.Peek@FIFA.org

## Abstract

Heading in football (soccer) is a complex skill involving deliberate head-to-ball contact, which may pose short-, medium-, and long-term risk to player brain health. However, understanding header exposure during matches and training sessions, as well as comparing header incidence between studies is currently challenging given the lack of standardisation in descriptors, definitions, and reporting methods. This Delphi study aimed to establish a consensus on the descriptors, definitions, and reporting methods for heading in football research to improve consistency and quality. The study involved 167 participants from diverse football-related backgrounds including coaches, players, medical personnel, and researchers, with consensus achieved to include 27 descriptors in minimum reporting criteria for heading in football research. An additional 27 descriptors were also defined for inclusion in an expanded framework. The operational definition of a header was standardised as "a head-to-ball contact where the player makes a deliberate movement to redirect the trajectory of the ball using their head." The consensus framework provides a standardised approach to heading in football research to enhance data quality and comparability across studies. Improved header incidence data quality has the potential to contribute significantly to our understanding of the risks associated with heading in football to inform future research and practice guidelines.

**Data availability statement:** All relevant data are within the paper and its Supporting information files.

**Funding:** This project was part funded by a Thompson Equity Prize (The University of Sydney) awarded to KP.

**Competing interests:** KP, AS, TM, TEA, VG are all currently employed by non-profit football organisations. AS and KP declare employment with FIFA. TM is chairman of UEFA´s and the German FA´s (DFB) Medical Committee as well as of the German League Organisation´s (DFL) working group "Medicine in Professional Football". TEA is Chief Medical Officer of the Football Association of Norway (NFF). VG is the Chief Medical Officer at FIFPRO. JG has been a paid contracted Injury Spotter for FIFA organised tournaments (2023). TM and VG are non-paid members of FIFA's Heading Expert Group. This does not alter our adherence to PLOS ONE policies on sharing data and materials. All authors declare no other relevant financial or non-financial competing interests.

## Introduction

Research on heading in association football, which includes the incidence and characteristics of headers, could be explored as one approach of assessing the potential causal inference between header exposure and long-term issues with brain health in retired players [1]. A recent systematic review and meta-analysis exploring the risk to football players in developing any neurodegenerative disease (including Alzheimer's disease and other dementias, motor neurone disease, amyotrophic lateral sclerosis, and Parkinson's disease) compared to general populations from nine included studies reported an odds ratio of 1.69 (95% CI 1.11–2.58) with significant heterogeneity between studies [2]. However, header, head impact, and/or head injury exposure was either not objectively collected or completely absent as an outcome measure in these studies, with causal inferences often based on career length [3], and playing position (out-field versus goal keeper) [1]. Causal inference frameworks assist in understanding causal relationships between factors particularly in circumstances where randomisation of exposure that ensures exchangeable comparison groups is considered infeasible [4]. The longitudinal collection of objective header data (for instance, using video analysis), based on a standardised framework of descriptors and definitions, could provide a more accurate report of header exposure, when compared with previous measures relying on player estimation [5]. Furthermore, objectively collected header exposure data can assist when exploring the potential short-term (defined as a single session of headers) [6], or medium-term (which includes header exposure over multiple practice sessions or across a season/s in active players) risk to player's brain health. Exploring the potential short- and medium-term risk associated with repeated headers offer an alternative study design given the infeasibility of completing a randomised clinical trial in which young players are assigned to play football with or without heading with their cognitive function monitored over many decades [7]. However, if heading in football research is to contribute to our understanding of the potential relationship between header exposure and short-, medium, or long-term brain health of football players then the collection of header exposure data needs to be of the highest quality possible.

A recent systematic review highlighted several shortcomings related to current heading in football research, most importantly the lack of a standardised operational definition of a header, with some studies making no distinctions between a deliberate head-to-ball contact (i.e., a header), and an accidental or unintentional ball-to-head impact (such as a player being hit in the face by a ball delivered at close range) [5]. This distinction is important given that higher head impact forces have been measured when the ball contact is unexpected [8]. Furthermore, there was an absence of minimum reporting criteria in these studies related to player demographics, heading descriptors and their definitions, as well as reporting methods [5]. This Delphi study aims to bridge this gap by establishing a framework of descriptors and definitions as well as reporting standards to improve the consistency and quality of heading in football research. Therefore, the objective of this study is to achieve consensus in the descriptors, definitions, and reporting options that should be included in minimum reporting criteria for heading in football research.

## Methods and analysis

### Project steering committee

A Project Steering Committee was convened that included prominent experts in heading in football research, and research methods, while also considering representation across professional backgrounds, research career stage, and country of origin. The final Project Steering Committee included 10 members, 4 women, including the project lead, and 6 men, who originated from 9 different countries worldwide. Professionally, 2 members were Delphi and consensus study experts, and 6 members had health, and health related research backgrounds. Two members were also former professional football players, and 1 has experience in coaching. Seven of the members were mid- to late-stage researchers and 3 were early-stage researchers, including 2 members completing PhDs which included heading incidence research. Collectively the committee have published more than 50 research papers on heading, and/or head injuries in football. See S1 Appendix for more details.

### Protocol, and ethics

The protocols for this study were registered on Open Science Framework (https://osf.io/qh6un and https://osf.io/qarxg) and was conducted using a phased approach, as described in Fig 1, following Guidance on Conducting and Reporting Delphi Studies (CREDES) [9], and ACcurate COnsensus Reporting Document (ACCORD) [10]. As this study did not involve the collection of personal health-related data, or other sensitive data, it was granted an exemption from ethics provided by Swiss Association of Research Ethics Committee, Switzerland (BASEC-Nr: Req-2O2 4-OO3.23).

### Systematic review

A systematic review was completed in December 2023 to collate the number and type of heading descriptors used in all published studies which report on heading incidence in football, as well as documenting the data collection and reporting methods used in the included studies to present heading incidence data. The review is published separately [5]. This review identified 71 studies which included header incidence data as an outcome measure for inclusion with the following key findings: 1) only 61% of studies defined a header with even fewer (23%) providing an operational definition of a header within the methods; 2) important study and player demographic data including year and country were often not reported; 3) reported heading descriptors and their reporting options varied greatly between studies; 4) visual identification of headers was essential when inertial measurement units were used to collect header incidence data; and 5) there was a lack of standardisation in the reporting methods used in header incidence studies making comparison between studies challenging, if not impossible [5]. All identified heading descriptors from the systematic review were included in the Phase 1 questionnaire.

For this study, heading was defined as a player performing a deliberate movement to redirect the trajectory of a ball using their head. Therefore, heading could result in either a header (with head-to-ball contact) or an attempted header (without head-to-ball contact) and could be performed in either a contested (duel) or uncontested (no duel) situation.

### Questionnaire development

Questionnaires were developed for each objective and phase of this project (Phase 1- descriptors, Phase 2- definitions, Phase 3- reporting methods). These questionnaires were distributed sequentially. In other words, once consensus was reached on the descriptors (Phase 1), a questionnaire to gain consensus on the definitions (Phase 2) was developed and distributed, followed by a questionnaire to gain consensus on the methods of data reporting (Phase 3), Fig 1. All questionnaires were developed and distributed using REDCap electronic data capture tools hosted by The University of Sydney. The Phase 1 first-round questionnaire also included a number of demographic questions

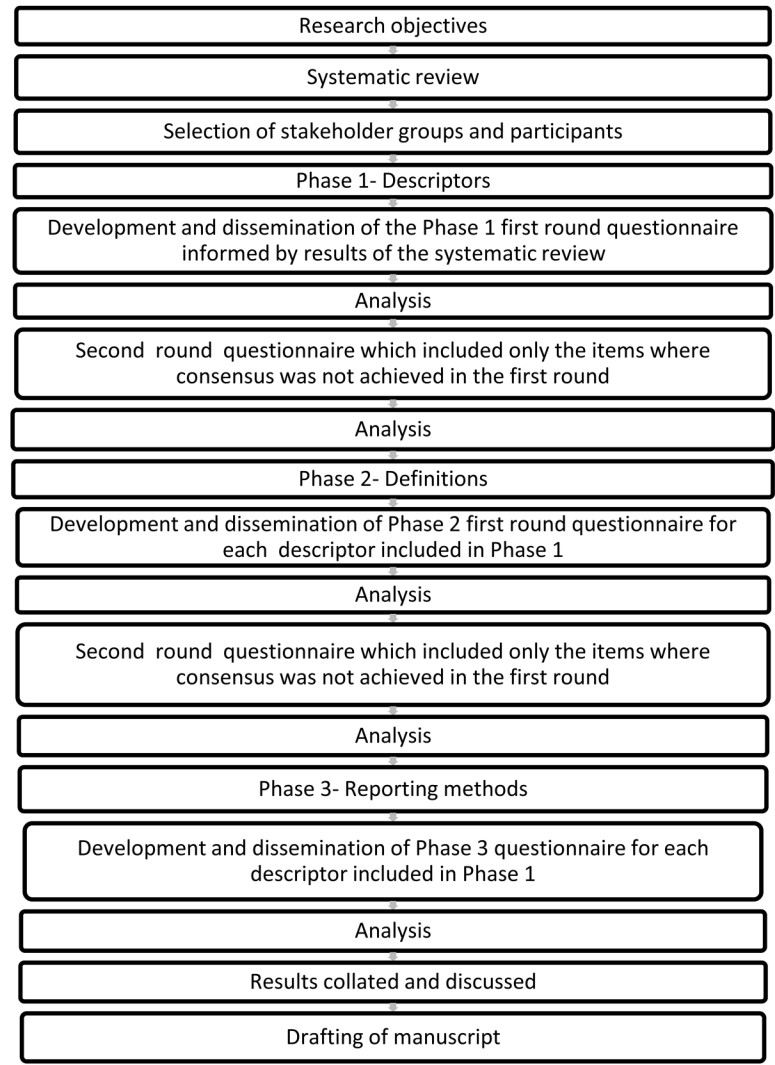

**Fig 1. Overview of the Delphi Method (adapted from Prinsen at al. [11]) for each phase 1-3 used in this study.**

to gain more information about each participant including: self-reported role in football (coach, medical personnel, player, researcher), age, gender, country where they had spent most of their football or working career, highest playing level (*players only*), gender and age group of players they mainly worked with (*coaches, medical personnel and researchers only*).

The following methods were applied to the development of each questionnaire.

**Phase 1 questionnaire.** Phase 1 included two questionnaires, first-round and second-round. These questionnaires focused on gaining consensus on the descriptors that should be included in minimum reporting criteria for heading in football research.

a) All descriptors included in the Phase 1 questionnaires were scored using options 1–9 where 1–3 was considered not important/do not include in minimum reporting criteria, 4–6 important but not critical to include, and 7–9 critical/definitely include in minimum reporting criteria.

b) Any descriptor scoring >80% agreement for options 7–9 (definitely include) in the first-round questionnaire was removed from the second-round questionnaire (as consensus to include had been reached) but were included in the Phase 2 (definitions), and Phase 3 (methods of reporting) questionnaires. These items are included in the final recommended list of descriptors to be included in minimum reporting criteria for heading in football research.

c) Any item scoring >80% agreement for options 1–3 (do not include) in the first-round questionnaire was removed from the second-round questionnaire and but were included in the questionnaires of Phase 2 (definitions) and Phase 3 (methods of reporting) to gain consensus on the definition and methods of reporting even though they were not recommended to be included in minimum reporting criteria for heading in football research.

d) Any items that were removed between rounds were clearly communicated to participants (including what and why). Participants had access to a 'free text' box to voice any concerns they had about any removed item in the second-round questionnaire.

e) The remaining items that did not reach consensus in the first-round questionnaire to be included, or excluded, from minimum reporting criteria were added to the second-round questionnaire,

f) Any descriptors suggested by any participant in the first-round questionnaire were also included in the second-round questionnaire. These additional descriptors were highlighted to participants as being suggested by a peer-participant.

g) To assist participants in completing the second-round questionnaire, participants were provided with the mean score for each descriptor (across all participants) as well as their previous score for that descriptor.

h) Descriptors that achieved >80% consensus at that point were automatically included in (or excluded from) minimum reporting criteria. Descriptors that did not achieve consensus (but scored >70%) were discussed and anonymously voted on by the Project Steering Committee as to whether to include, or exclude, and reported accordingly in subsequent questionnaires.

**Phase 2 questionnaire.** Phase 2 included two questionnaires, one in a first-round and the other in a second-round. These questionnaires focused on achieving consensus on the definition of each descriptor included in Phase 1.

a. In the Phase 2 questionnaires, participants were provided with three lists of descriptors:

   i. The first list contained the descriptors that reached >80% consensus in Phase 1 to be included in minimum reporting criteria for heading in football research as scored by the participants in Phase 1.

   ii. The second list contained the descriptors that did not reach consensus in Phase 1 but were voted for inclusion in minimum reporting criteria by the Project Steering Committee.

   iii. The third list contained the remaining descriptors that did not reach consensus by participants in Phase 1 or the Project Steering Committee but were included in this questionnaire to gain a standardised definition for these descriptors (for potential use in future research).

b. Below each descriptor, participants were provided with one or more definitions as reported in the published literature (extracted from the published systematic review) [5] to define or describe that descriptor. In cases where no published definition was available, a suggested definition was developed by the Project Steering Committee.

c. Participants were instructed to either select their preferred definition, or to propose a definition of their own.

d. Any preferred definition scoring >80% agreement in the first-round questionnaire (and without a suggested alternative definition) was removed from the second-round questionnaire (as consensus to include had been reached).

e. The remaining descriptors without >80% agreement on the preferred definition were included in the second-round questionnaire, along with any alterative definition/s suggested by any participant in the first-round questionnaire where available. These alternative definitions were highlighted to participants in the second-round questionnaire as being suggested by their peers.

f. To assist participants in completing the second-round questionnaire, participants were provided with the mean results (percentage) on the preferred definition of each descriptor as well as their previously selected preferred definition.

g. Descriptors that reached >80% agreement on the preferred definition were copied over to the Phase 3 questionnaire.

**Phase 3 questionnaire.** Phase 3 only included one questionnaire (although a second-round questionnaire was possible, if needed). This questionnaire focused on the methods and reporting options for each descriptor from Phase 1.

a. In Phase 3 participants again were provided with the three lists of descriptors as per the Phase 2 questionnaire.

b. Below each descriptor, participants were provided with the preferred definition as per the results of the Phase 2, as well as the reporting options and methods of reporting for each descriptor. Methods and reporting options were identified from the included studies as part of the systematic review [5]. In cases where no published definition was identified, methods and reporting options were developed by the Project Steering Committee.

c. Participants were asked if they agreed with the methods and reporting options for each list of descriptors with an open text box for participants to add their comments or feedback where they felt edits or amendments should be made.

d. Where >80% of participants agreed with the methods and reporting options in Phase 3, these were included in the final framework of recommended descriptors, definitions, and reporting methods to be used in heading in football research.

**Selection of stakeholder groups and participants.** The selection of stakeholder groups from which participants were derived, were discussed, and selected by the Project Steering Committee to reflect the wider football population that might conduct heading in football research or review the results of such studies. The final selected stakeholder groups included:

• Researchers

• Medical personnel (from varying professional disciplines including team physicians and physiotherapists),

• Technical and performance coaches and analysts, and

• Players (including past and present players)

All participants needed to be 18 years or older to participate. Researchers were selected based on the corresponding author of the included studies in the published systematic review [5]. Medical personnel, coaches and players were recommended by members of the Project Steering Committee based on their networks and knowledge of who could be considered as having expertise on heading in football. Diversity in terms of confederation, country of origin, sex, and ethnicity of participants was also considered. Prior to the distribution of the first questionnaire all potential participants (except researchers as their email addresses were publicly available) were emailed information about this project by the member of the Project Steering Committee who recommended them, with only those agreeing to participate receiving the first questionnaire in Phase 1.

**Questionnaire distribution timeframe.** The first questionnaire was distributed on 1st of May 2024, with the fifth and final questionnaire distributed on 29th November 2024, closing on 23rd December 2024. Each questionnaire was open for four weeks with two reminders sent to non-responders.

**Completion rate and missing data.** The final list of participants as confirmed by the Project Steering Committee were emailed the first-round questionnaire in Phase 1. Any participant who did not complete this first questionnaire (or declined to participate) were removed from further involvement in the project. Although participants were encouraged to complete

every questionnaire, only completion of the Phase 1 first round questionnaire was required to remain involved in this study. Records of completion rate and any missing data were recorded and reported below (see also S1 Appendix for the complete dataset for each questionnaire).

## Results

### Demographic and completion data

Initially 244 people were invited to participate including researchers (n=66), medical personnel (n=79), coaches (n=58), and players (n=41). Seventeen emails were returned as undeliverable (all researchers), attempts were made to locate alternative email addresses where able. Three people declined to participate (one coach, one medical personnel, and one researcher) citing time-related factors as a reason to decline. No response was received from a further 57 people. Non-responders included 31 coaches, 11 medical personnel, 8 players and 7 researchers. Of the non-responders, 46 were men and 14 were women, with 15 from Confederation of African Football (CAF), 6 from Confederation of North, Central America and Caribbean Association Football (CONCACAF), 1 from Oceania Football Confederation (OFC), and 38 from Union of European Football Associations (UEFA).

This resulted in 167 participants completing the first questionnaire (68% of invitees). In total, 18 participants (11%) only completed the first questionnaire. All other participants completed a minimum of two questionnaires, with a total of 88 participants (53%) completing all five questionnaires. The highest retention rate of participants were coaches (69%) and medical personnel (64%). Participant demographic and completion data are reported in Table 1.

Two-thirds (n=111, 66%) of participants were men with a mean age of 41.7 years, with approximately half (n=66, 49%) of coaches, medical personnel, and researchers (n=134) reporting that they mainly work with boys/men players (n=26, 19% reported that they mainly work with girls/women players, with the remaining n=42, 32% working with both boys/men and girls/women). Only 4 (3%) coaches, medical personnel and researchers worked mainly with children (12 years or younger, these participants were all coaches), 37 (28%) worked with adolescents (13–17 years), and 93 (69%) worked with adult players. Two medical personnel worked in para-football (one physiotherapist and one doctor).

Participants also played or worked in a range of geographical locations and confederations; Europe (UEFA, n=82), Africa (CAF n=35), North America, Central America and the Caribbean (CONCACAF n=31), Asia (Asian Football Confederation (AFC) n=9), Oceania (OFC n=8), and South America (South American Football Confederation (CONMEBOL) n=2). Of the 33 players who completed the first questionnaire, 32 (97%) were professional players.

### Phase 1- Descriptors

The Phase 1 first-round questionnaire included 46 descriptors, 10 of which reached >80% consensus as essential to be included in minimum reporting criteria for heading in football research, particularly where it relates to header incidence and characteristics during match play or training sessions. These were:

• Activity type

• Playing level

• Number of players (overall)

• Number of teams

• Sex

• Age

• Age group

**Table 1. Participant demographic and completion data.**

| Role | Men (M); or Women (W) (n,%)^ | Mean (SD) age in years | Mainly work with boys/men (M); or girls/women (W); or both (B) (n,%) | Mainly work with players aged ≤17 years; or ≥18 years (n,%) | Completion rate Q1 (n) | Completion rate Q2 (n,%) | Completion rate Q3 (n,%) | Completion rate Q4 (n,%) | Completion rate Q5 (n,%) | Completion rate Q1-Q5 (n,%) |
|---|---|---|---|---|---|---|---|---|---|---|
| **Coach** N=26 (16%) | 20, 77% (M); 6, 23% (W) | 45.1 (11.4) | 15, 58% (M); 5, 19% (W); 6, 23% (B) | 15, 58% (≤17); 11, 42% (≥18) | 26 | 25, 96% | 21, 81% | 19, 70% | 19, 73% | 18, 69% |
| **Medical personnel** N=67 (40%) | 49, 73% (M); 18, 27% (W) | 44.7 (13.9) | 35, 52% (M); 12, 18% (W); 20, 30% (B) | 7, 10% (≤17); 60, 90% (≥18) | 67 | 62, 93% | 57, 92% | 51, 76% | 54, 81% | 43, 64% |
| **Player** N=33 (20%) | 15, 45% (M); 18, 55% (W) | 34.8 (7.3) | N/A | N/A | 33 | 26, 79% | 21, 63% | 14, 42% | 19, 58% | 12, 36% |
| **Researcher^** N=41 (25%) | 27, 68% (M); 13, 31%(W) | 46.3 (13.2) | 16, 39% (M); 9, 22% (W); 16, 39% (B) | 19, 46% (≤17); 22, 54% (≥18) | 41 | 31, 76% | 22, 54% | 18, 45% | 23, 56% | 15, 37% |
| **Total=167** | 111, 66%(M); 55, 33% (W) | 41.7 (11.5) | 66, 49% (M);* 26, 19% (W); 42, 32% (B) | 41, 31%(≤17);* 93, 69% (≥18) | **167, 100%** | **144, 86%** | **121, 72%** | **102, 61%** | **115, 69%** | **88, 53%** |

*total 134 (minus 33 players where this question was not applicable) ^ one person preferred not to state (total n =166).

- Type of header

- Head injury

- Non-header related head impact

In addition, three descriptors reached >80% consensus to not be included in minimum reporting criteria for heading in football research. These were:

- Ball deliverer

- Match score at the time the header occurred

- Final score

Participants also suggested the following additional descriptors, which were then added to the second-round questionnaire:

- Head covering or head personal protective equipment: The player performing the header was observed to be wearing something that covered all or part of their head or face at the time of the header (such as a hijab) or a form of personal protective equipment (such as head gear, or face mask/shield)

- Head movement: The movement direction of the player's head immediately before and/or after they completed the header

- Head impact magnitude: The linear and/or rotational acceleration forces associated with the header

- Weather: The weather at the time of the header (i.e., raining, snowing, sunny)

- Ball: Information about the ball being used

- Ball speed: The speed of the ball immediately before and/or immediately after the header

- Referee: The action of the referee in response to the heading event (such as foul called, signal for medical team, play on, VAR)

- Player size: The relative size of the player performing the header (such as smaller, larger or similar in size to the other players on the pitch)

- Player stability: How balanced the player heading the ball was at the time of the header

- Ball tracking: Whether the player was tracking the in-coming ball immediately prior to the header.

- Playing surface: The specific playing surface of the pitch (i.e., grass, artificial turf)

After completion of the Phase 1 second-round questionnaire, a further 14 descriptors were included as minimum reporting criteria for heading in football research. Five descriptors scored over 70%, which were then discussed by the Project Steering Committee, who then voted for their inclusion in minimum reporting criteria via completion of an anonymous questionnaire (three of which were then added to the minimum reporting criteria). Thus, a total of 27 descriptors were determined to form the minimum reporting criteria (Table 2, Column 1).

**Phase 2 Definitions**

For the Phase 2 questionnaire, 3 lists were created. Forty-two out of 54 descriptors attained >80% consensus on their preferred definition after the first-round questionnaire, with 8 of the 12 remaining descriptors having at least 1 alternative definition suggested by a participant in the first-round questionnaire. These 12 descriptors were then included in the

**Table 2. Descriptors with >80% consensus to be included in minimum reporting criteria for heading in football research with their definitions and reporting options.**

| Descriptor | Definition | Reporting options |
|---|---|---|
| **Demographic descriptors** | | |
| Data collection dates | The date, duration, or period that data were collected. | To be reported as open text which states the specific date (in day, month and year) when data collection commenced and finished. |
| Country: | The country or countries where data were collected. | To be reported as open text which states specific name of the country or countries. |
| Activity: | The activity type that the data relates to. *(NB additional open text information about the match or training session can also be included as required)* | To be reported using the following options:<br>i. Match: A game of football played under competition rules<br>ii. Football specific training: A training session or drill (includes small-sided games during training) |
| Competition, tournament or league: | The specific name of the competition, tournament or league where the data were collected (where applicable). | To be reported as open text which states the specific name of the competition, tournament or league |
| Playing level: | The playing level of the players that the data relates to. To be reported using the following options: | To be reported using the following options:<br>i. Professional: a player who has a written contract with a club and is paid more for his/her footballing activity than the expenses he/she effectively incurs.<br>ii. Amateur: a player who plays organised football but does not meet the criteria of a professional<br>iii. Recreational: a player who plays non-organised football |
| Number of players (overall): | The total number of players for whom data were collected (including number of players included in any sub-group analyses). | To be reported as open text which states specific number of players for whom the data relates. |
| Number of teams: | The number of teams for whom data were collected. | To be reported as open text which states the specific number of teams that the data relates to. |
| Sex: | The biological sex of the players for whom data were collected. | To be reported using the following options:<br>i. Male: male at birth<br>ii. Female: female at birth<br>iii. Other: Sex at birth was reported as a term other than male or female<br><br>NB as an alternative this could be reported by the competition in which the player is competing:<br>i. Men or boys<br>ii. Women or girls<br>iii. Mixed |
| Age: | The age of the players for whom data were collected at the time of data collection. | To be reported as open text which states the mean age plus range and standard deviation (in years). |
| Age group: | The age group or groups that the players compete in for whom data were collected. | To be reported as open text which states the age group as per league/competition/tournament rules (such as under-9s, under-18s, senior) |
| **Heading descriptors** | | |
| Type of header: | The type of header that is performed. | To be reported using the following options:<br>i. Block – A player heads the ball in an attempt to stop the opposition's in-possession action (i.e., pass, cross, or shot on goal) reaching its intended target without the aim of retaining possession of the ball for themselves.<br>ii. Clearance – A player decides to clear the ball up field or out of play, attempting to relieve pressure on their team using their head with their first touch.<br>iii. Interception – The player heading the ball appears to anticipate where the ball is being distributed, the ball is headed with the intention of winning possession for themselves or their team.<br>iv. Attempt at goal: – A header is performed by a player with the intention of scoring a goal.<br>v. Pass – A player has used their head to play a pass which is defined as a distribution action performed by a player and a teammate with the intention of keeping possession of the ball.<br>vi. Self-serve – The player tries to gain control over the ball themselves.<br>vii. Unclear- Unable to accurately report. |

*(Continued)*

**Table 2.** (Continued)

| Descriptor | Definition | Reporting options |
|---|---|---|
| Controlled or uncontrolled header: | The level of control in the intentional redirection of the ball that is demonstrated by the player performing the header. | To be reported using the following options:<br>i. Controlled header: Where a player demonstrates a level of control in the redirection of the ball (for example, if a player redirects the ball towards goal or towards a teammate).<br>ii. Uncontrolled header: Where the player does not demonstrate a level of control in the redirection of the ball (for example, the ball bounces up in the air or goes in a direction that does not advantage the player performing the header).<br>iii. Unclear: Unable to accurately observe/report variable. |
| Head Injury: | The header resulted in a potential head injury or diagnosed head injury. | To be reported using the following options (NB more than one option might apply):<br>i. Potential head injury where a player heading the ball goes to ground for 5 seconds or longer before resuming play.<br>ii. Potential head injury where a player heading the ball receives on-pitch medical assessment.<br>iii. Potential head injury where a player heading the ball receives off-pitch medical assessment.<br>iv. Concussion substitution is used.<br>v. Diagnosed head injury verified by the player or team.<br>vi. Unclear: Unable to accurately observe/report variable. |
| Non-header related head impact: | A head impact that is not classified as a header. | To be reported using the following options (NB more than one option might apply):<br>i. Unintentional ball-to-head impact: any physical contact between a player's head and the ball, which is not considered deliberate (for example, ball impact with a player's face or head when the ball is played at short range or ball impact to the back of a player's head).<br>ii. Head impact – other player: any physical contact between a player's head or face and another player's body part (for example, head-to-head, elbow-to-head impact)<br>iii. Head impact – object: any physical contact between a player's head or face and an object (such as ground-to-head, goalpost-to-head impact). |
| Ball-to-head point of contact: | The area of the head where the ball makes contact during the header. | To be reported using the following options (NB more than one option might apply when the ball contact area crosses one or more areas):<br>i. Forehead: The area between the eyebrow ridge and hairline<br>ii. Top of head: The area between the hairline and crown of head<br>iii. Side of head: The area on the lateral surfaces of the cranium inclusive of ears<br>iv. Back of head: The area on the posterior surface of the cranium, posterior to the crown of the head.<br>v. Face: All areas of the face inferior to the eyebrow ridge.<br>vi. Unclear: Unable to accurately observe/report variable. |
| **Ball Descriptors** | | |
| Ball distance before header: | The distance the ball travelled before it was headed. | If the exact number of metres that the ball travelled before a header can be calculated, then this should be reported either as open text in metres (along with the tool used to calculate ball distance) or distance can be reported using the following options:<br>i. <10m (short): Ball travelled a short distance (<10m) prior to header.<br>ii. 10-30m (medium): Ball travelled a medium distance (10-30m) prior to header.<br>iii. >30m (long): Ball travelled a long distance (>30m) prior to header.<br>iv. Unclear: Unable to accurately report. |
| **Player descriptors** | | |
| Playing position: | The tactical or field position that the player heading the ball was in at the time of the header. | To be reported using the following options:<br>i. Goalkeeper<br>ii. Central defenders<br>iii. Full backs<br>iv. Wingers<br>v. Midfielders (includes defensive, attacking, central and other midfielders)<br>vi. Strikers/ forwards |

*(Continued)*

| Descriptor | Definition | Reporting options |
|---|---|---|
| Player movement: | The movement of the player immediately prior to the header. | To be reported using the following options:<br>i. Jumping: Player who headed the ball was jumping immediately prior to the header. This includes attempting/preparing to jump, in the air following a jump, or landing from a jump.<br>ii. Stationary: Player who headed the ball was standing bipedally, such that the position of their feet did not move prior to performing the header. This includes upright stance and athletic/squat stance.<br>iii. Walking: Player who headed the ball engaged in locomotion with at least one foot remaining in contact with the ground. The player must have completed a minimum of one stride (from touchdown of one foot to the next touchdown of the same foot) prior to performing the header.<br>iv. Running: Player who headed the ball engaged in locomotion with continuous and repetitive steps including a flight phase in which both feet are above the ground. The player must have completed a minimum of one stride (from touchdown of one foot to the next touchdown of the same foot) prior to performing the header. This includes accelerating, decelerating, and running at a steady speed.<br>v. Positioning movements: Player who headed the ball produced small movements with their feet immediately prior to the header to adjust their body in the optimal position to perform the header. This includes shuffling movements with the feet which do not reflect natural gait, single steps which do not constitute a complete stride, and turning/pivoting their body on the spot to face the ball.<br>vi. Diving: Player who headed the ball was intentionally diving, such that their body was parallel to the ground prior to performing the header.<br>vii. Falling: Player who headed the ball loses balance/stability and ends up in a non-upright position on the ground prior to performing the header.<br>viii. Sliding: Player who headed the ball was intentionally sliding prior to performing the header.<br>ix. Unclear: Unable to accurately report. |
| Heading under pressure: | Whether the player completing the header was under direct, indirect or no pressure from an opponent player at the time of the header. | To be reported using the following options:<br>i. Direct pressure: Player who headed the ball was under clear pressure from an opponent who is actively attempting to disrupt or contest for possession. Any body contact between the attacker and the defender also equates to direct pressure. This includes all in-contest (duel) situations.<br>ii. Indirect pressure: Player who headed the ball was under pressure from an opponent trying to control the direction in which the ball can be played. The opposition player is not within an appropriate distance to compete for possession and is not attempting to directly recover possession themselves.<br>iii. No pressure: Player who headed the ball was under no pressure and was able to perform an action freely in any direction.<br>iv. Unclear: Unable to accurately report |
| Contested nature of the header: | The header was performed during a contested (duel) or uncontested (no duel) situation. | To be reported using the following options:<br>i. Contested – Two or more opposing players were observed to compete for the header in close enough proximity that physical contact between players was possible (i.e., within one arm's length distance).<br>ii. Uncontested – No opposing player was observed to compete for the ball, or the nearest opposition player was far enough away that no physical contact between players was possible. |
| Aerial duel: | Two or more players competing for a ball that is above shoulder height; where at least one player is off the ground and is being physically challenged by an opposition player | To be reported using the following options:<br>i. Yes: A contested header where at least one of the players was off the ground.<br>ii. No: All players competing for the ball were grounded. |
| Physical (header) duel: | Two or more players physically competing for a header and with all the players involved being grounded. | To be reported using the following options:<br>i. Yes: A contested header where all the players were grounded.<br>ii. No: A contested header where at least one player was not grounded. |

*(Continued)*

**Table 2.** (Continued)

| Descriptor | Definition | Reporting options |
|---|---|---|
| Body contact during a header: | A situation where two or more players are competing for the header (aerial or physical duel) and the nature of the body contact observed between players. | To be reported using the following options (NB more than one option might apply):<br>i. Head-to head contact<br>ii. Body-to-body contact (includes any non-head related body contact between players such as trunk-to-trunk contact)<br>iii. Shoulder/upper arm-to-head contact<br>iv. Elbow-to-head contact<br>v. Forearm/wrist/hand-to-head contact<br>vi. Trunk/pelvis-to-head contact<br>vii. Thigh-to-head contact<br>viii. Knee/lower leg-to-head contact<br>ix. Foot/ankle-to-head contact<br>x. No body contact between players |
| **Match descriptors** | | |
| Consequence/outcome of the header: | The immediate outcome of the header. | To be reported using the following options:<br>i. Possession won: Immediately following a defensive header (i.e., clearance, block, interception) or a contested header, the team of the player heading the ball wins possession of the ball if the final touch before the header comes from the opposition team, and: i) the next touch of the ball is made by a player on the heading player's team; or ii) if the referee sanctions an opponent for committing a foul against the heading player during the header, resulting in the heading player being awarded a free kick.<br>ii. Possession lost: Immediately following a distribution header (i.e., pass, cross) or a contested header, the team of the player heading the ball lose possession of the ball if the final touch before the header comes from a player on the heading player's team, and: i) the next touch of the ball is made by a player from the opposition team; ii) the ball goes out of play; or iii) the referee sanctions the heading player for committing a foul against an opponent during the header, resulting in the opponent being awarded a free kick.<br>iii. Possession retained: Immediately following a distribution header (i.e., pass, cross) or a contested header, the team of the player heading the ball retain possession of the ball if the final touch of the ball before the header and the next touch of the ball after the header comes from a player on the heading player's team.<br>iv. Opposition retained: Immediately following a defensive header (i.e., clearance, block, interception) or contested header, the opposition team retain possession of the ball if the final touch of the ball before the header and the next touch of the ball after the header comes from a player on the opposing team to the player heading the ball.<br>v. Goal scored: Following the player's headed attempt at goal, the ball passes the goal line within the goal frame. The heading player is awarded the goal for their team and play restarts from the centre circle.<br>vi. Own goal: The player heads the ball in their defensive area, and the ball passes the goal line within the goalframe they are defending. The opposition team is awarded the goal.<br>vii. Goal attempt – On target: Following the player's headed attempt at goal, the ball is on trajectory to end up within the goal frame, though does not result in a goal due to being prevented by a defensive/goalkeeping action or obstructed by the body of a player.<br>viii. Goal attempt – Off target: Following the player's headed attempt at goal, and without any external influences on its trajectory – the ball does not finish inside the goal frame. This includes the ball hitting the goal frame and not crossing the goal line.<br>ix. Goal – disallowed: Following the player's headed attempt at goal, the ball passes the goal line within the goal frame. The subsequent goal is i) not awarded to the heading player's team due to being ruled offside; or ii) revoked due to a sanction that occurred in the build-up to the goal.<br>x. Unclear: Unable to accurately report. |

*(Continued)*

**Table 2.** (Continued)

| Descriptor | Definition | Reporting options |
|---|---|---|
| **In addition, the following three descriptors did not reach >80% consensus but were voted for inclusion in minimum reporting criteria by the Project Steering Committee:** | | |
| National or international: | Whether data were collected during a national or international tournament or competition. | To be reported using the following options:<br>i. International: A tournament or competition or friendly played between teams from two or more countries (to include international league ranking -where accessible)<br>ii. National: A league, tournament, competition or friendly played within one country (to include playing division -out of the total number of divisions in the league system – where accessible) |
| Number of players per activity type: | The number of players per team (for match data) or the number of players involved in each practice session or training activity. | To be reported as open text which states the specific number of players. |
| Drill (for practice sessions or training activities only): | The specific activity or drill the player was involved in at the time of the header. | To be reported as open text which states the specific name or aim of the activity or drill. |

second-round questionnaire along with the 8 alternative definitions. Following completion of the second-round questionnaire, all 54 descriptors had achieved >80% consensus on the preferred definition (including three descriptors which were suggested by a participant in the first-round: Time between headers, Head impact magnitude, and Player size).

## Phase 3 Methods and reporting options

All descriptors in the Phase 3 questionnaire reached >80% agreement for the final definition, methods and reporting options, except for 2: 'head injury', and 'body contact during a duel/header'. For the 'head injury' descriptor, 78% of participants selected the definition "the header resulted in a potential head injury or diagnosed head injury," the other 4 definitions ranged from 2–13%. For the 'body contact during a duel/header' descriptor 78% of participants selected "a situation where two or more players are competing for the header and the nature of the body contact observed between players" definition, the other 2 definitions ranged from 6–16%.

There were 5 comments from participants that suggested minor wording clarification of 2 reporting options. These were for the descriptor of 'Playing level' in list one where the addition of "within a national league system" was recommended to the 'amateur' reporting option, as well as the descriptor 'Referee action' in list 3 where the option "no referee action" was suggested by 3 participants. As these were minor suggestions, these were discussed by the Project Steering Committee only. Five additional comments were received from participants to outline that they did not agree with the descriptor being included in either that list or being collected at all, no changes were made based on these comments (as consensus had already been achieved across the 3 Phases) although these comments were noted. Two further comments were made regarding being able to reliably collect data for a number of descriptors, again no changes were made as this was beyond the scope of this project.

**Operational definition of a header and heading incidence reporting.** In Phase 3, the operational definition of a header was determined to be:

• A head-to-ball contact where the player makes a deliberate movement to redirect the trajectory of the ball using their head.

It was noted that minimum reporting for heading in football research should include raw number of headers per activity as well as incidence rate per 1000 match (or training) hours. Where data are collected from both matches and training sessions, these data should be reported separately with the exposure hours used to calculate the incidence rate also being reported.

**Descriptors, definitions and reporting options**

The final descriptors, definitions and reporting options that reached >80% consensus to be included in minimum reporting criteria for heading in football research can be seen in Table 2. Descriptors that did not reach >80% consensus to be included in minimum reporting criteria but were included to gain a standardised definition and reporting options for these descriptors are reported in Table 3. See S1 Appendix for a full list of results.

## Discussion

The aim of this Delphi project was to gain consensus on a list of descriptors, their definitions, and reporting options that should be included in minimum reporting criteria for heading in football research, particularly research related to header incidence and characteristics during match play and/or training sessions. This resulted in 27 descriptors being recommended, including 24 descriptors which achieved consensus from participants across a range of relevant footballing backgrounds including coaches, medical personnel, players, and researchers. An additional 3 descriptors that reached >70% and were added to the minimum reporting criteria by the Project Steering Committee. A further 27 descriptors also achieved agreed definitions, and reporting options by participants, thereby expanding the framework of standardised descriptors to 54. Having an expanded framework provides increased flexibility when collecting heading data, where additional descriptors beyond the minimum recommended can be collected in a standardised format to address the varied and specific research aims and objectives of future studies.

While having 27 recommended descriptors as minimum reporting criteria for heading in football research may seem a lot, 12 of these descriptors are categorised as demographic descriptors which should only require collection as summary data (such as data collection dates, country (or countries) where data were collected, and number, playing level, sex, and age of players included in the research). Of the remaining 15 descriptors, 7 relate to the player heading the ball, 5 relate to the header itself, 2 relate to the activity type, and the last 1 relates to the ball. Many of the recommended descriptors that relate to the player heading the ball, as well as the header itself, are important to capture as they can assist in understanding the potential risk posed by headers, and the action of heading, as opposed to non-header related head impacts.

One of the most fundamental findings from the systematic review completed to inform this Delphi study was that there were inconsistencies in the operational definition of a header (where one was provided) [5]. We now recommend that a header is defined as a "head-to-ball contact where the player makes a deliberate movement to redirect the trajectory of the ball using their head". This is considered important to assist in differentiating between headers, which are a deliberate sport specific action, and head impacts which are usually unintentional, and may pose a different risk to brain health.

While a number of studies have reported on the potential increased risk posed to former football players in developing a neurodegenerative disease [1,2,12], incidence data related to headers and head impacts have not been reported. One recent study of 199 former players (aged 50 years and older) recruited through the English Professional Footballers' Association, which included an estimation of incidence data, reported no overall association between headers, or other non-header related head impacts, and cognitive function, except possibly in forward players who were exposed to higher numbers of non-heading related head impacts [13]. Given that forward players have consistently been shown to head the ball less than other playing positions, particularly central defenders [14], underscores the importance of categorising headers separately from head impacts. In our study, participants recommended that non-header-related head impacts are defined as "a head impact that is not classified as a header", with reporting options including unintentional ball-to-head impacts, head impacts with another player, and head impacts with an object.

Further, capturing injury data that separate the mechanism of injury between those caused by headers, and those caused by head impacts is vital, given that headers are a less common cause of a potential head injury than head impacts [15]. While players can be injured by the ball, this is more commonly as a result of unintentional ball-to-head impacts [16], with reportedly higher rates of ball-related head injuries in women and girls [17]. An additional finding from the earlier study was an association between self-reported concussion and lower cognitive function using the Preclinical Alzheimer

**Table 3. Descriptors that did not reach >80% consensus to be included in minimum reporting criteria for heading in football research but were included in Phase 2 and 3 to gain consensus on their definitions and reporting options.**

| Descriptor | Definition | Reporting options |
|---|---|---|
| **Heading descriptors** | | |
| Attempted header: | A player attempting to head the ball without contact being observed between the head and the ball (this includes the other player/s in a contested situation). | To be reported using the following options:<br>i. Attempted header in an uncontested situation<br>ii. Attempted header in contest<br>iii. No attempted header observed<br>iv. Unclear- Unable to accurately report. |
| Time between headers: | The time that has elapsed between headers from the same player. | To be reported as open text which states the time (in minutes and seconds) between headers. |
| **Pitch descriptors** | | |
| Pitch location: | The location on the pitch where the header occurred. | To be reported using the following options:<br>i. Defensive third: Header was performed in the defensive third of the pitch.<br> a. Defensive penalty area (18-yard/16.5m box)<br> b. Defensive goal area (6-yard box/5.5m box)<br>ii. Middle third<br> a. Own half<br> b. Opponent half<br>iii. Attacking third<br> a. Attacking penalty area (18-yard/16.5m box)<br> b. Attacking goal area (6-yard/5.5m box)<br>iv. Unclear: Unable to accurately report. |
| **Ball Descriptors** | | |
| Ball distance- after header: | The distance the ball travelled after it was headed until the ball either hit the ground, an object, went out of play, scored a goal or was touched by another player. | If the exact number of metres that the ball travelled after the header can be calculated, then this should be reported either as open text in metres (along with the tool used to calculate ball distance) or distance can be reported using the following options:<br>i. <10m (short): Ball travelled a short distance (<10m) after the header.<br>ii. 10-30m (medium): Ball travelled a medium distance (10-30m) after the header<br>iii. >30m (long): Ball travelled a long distance (>30m) after the header.<br>iv. Unclear: Unable to accurately report. |
| Ball delivery type/ situation: | The delivery of the ball prior to header being performed. | To be reported using the following options:<br>i. Pass: A distribution action performed by a player with the intention of keeping possession of the ball. A player can manoeuvre the ball on the ground or aerially between themselves and a team-mate.<br>ii. Cross: A distribution action performed by a player with the intention of creating a goalscoring opportunity. The player can play the ball on the ground or aerially from any crossing zone with the intention of finding a team-mate inside the recognised target area.<br>iii. Clearance: An action where a player attempts to clear the ball upfield or out of play, usually to relieve the pressure or danger faced by themselves or their team.<br>iv. Attempt at goal: A distribution action performed by a player with the intention of scoring a goal.<br>v. Loose ball: The ball was loose with no team in controlled possession of the ball prior to the header.<br>vi. Free kick: The stationary ball is played from the position where the free kick was awarded to restart play.<br>vii. Corner – direct into box: The stationary ball is played from the corner quadrant to restart play. The ball is delivered directly into the penalty area.<br>viii. Corner – other: The stationary ball is played from the corner quadrant to restart play. The ball is not delivered directly into the penalty area. This includes short corners, and combination play.<br>ix. Goal kick: The goalkeeper (or defender) plays the stationary ball from the 6-yard line after the ball has gone out of play behind the goal line by the opposing team to restart play.<br>x. Goalkeeper kick from ground (open play): The goalkeeper distributes the ball by kicking the ball from the ground during open play.<br>xi. Goalkeeper kick from hands (punt): The goalkeeper distributes the ball by kicking the ball from their hands.<br>xii. Goalkeeper throw: The goalkeeper distributes the ball by throwing it.<br>xiii. Throw in: The ball is thrown overhead with both hands from the touchline to restart play.<br>xiv. Rebound: A player receives the ball following the event of either an attempt at goal which is saved and deflected by the Goalkeeper, blocked off the line or strikes the goal frame.<br>xv. Own ball: A player receives the ball directly after their attempted distribution of the ball is blocked or interrupted.<br>xvi. Unclear: Unable to accurately report. |

*(Continued)*

**Table 3.** (Continued)

| Descriptor | Definition | Reporting options |
|---|---|---|
| Ball: | Information about the ball being used in the match or training session. | To be reported as open text which may include the ball size, pressure, make, and model. |
| Ball deliverer: | The position and team of the player who played the ball immediately before the header. | To be reported using the following options:<br> i. Goalkeeper – Player plays in goal- own team.<br> ii. Goalkeeper – Player plays in goal- opposition team.<br> iii. Central defender – own team.<br> iv. Central defender – opposition team.<br> v. Full back-own team<br> vi. Full back – opposition team<br> vii. Winger- own team<br> viii. Winger- opposition team<br> ix. Midfielder – Includes defensive, attacking, central and other midfielders- own team.<br> x. Midfielder – Includes defensive, attacking, central and other midfielders- opposition team.<br> xi. Forward/striker – own team.<br> xii. Forward/striker – opposition team. |
| Angle of ball direction: | The angle of flight of the ball immediately before it is headed. | To be reported using the following options:<br> i. $< 0$ degrees (ball was in the upswinging phase of flight)<br> ii. 0–45 degrees<br> iii. $> 45$ degrees |
| Ball speed: | The speed of the ball immediately before and/or immediately after the header (if available). | To be reported as open text which includes the ball speed and how this was measured (including make and model of measurement device and measurement accuracy data). |
| **Player (or team) descriptors** | | |
| Attack or defence: | The team of the player heading the ball were in an attacking phase or defensive phase at the time of the header. | To be reported using the following options:<br> i. Attacking phase: The team of the player heading the ball has possession and is trying to build up an attack, create shooting opportunities, and score.<br> ii. Defensive phase: The team of the player heading the ball does not have possession of the ball and is trying to regain it, prevent the opponent from progressing, and protect the goal.<br> iii. In contest: No team has clear possession of the ball prior to the header.<br> iv. Unclear: Unable to accurately report. |
| Possession: | Whether the team of the player performing the header was in possession, out of possession or in contest, when the header was performed. | To be reported using the following options:<br> i. In possession: The team of the player who headed the ball was in controlled possession of the ball prior to the header.<br> ii. Out of possession: The team of the player who headed the ball was not in controlled possession of the ball prior to the header.<br> iii. In contest: Neither team has controlled possession of the ball prior to the header.<br> iv. Unclear: Unable to accurately report |
| Elbow position: | The elbow position of the player when performing the header. | To be reported using the following options (NB more than one option might apply if the player's elbows are in different positions – therefore left and right elbow position may need to be recorded separately):<br> i. Below shoulder height<br> ii. At shoulder height<br> iii. Above shoulder height<br> iv. Unclear: Unable to accurately report |

*(Continued)*

**Table 3.** (Continued)

| Descriptor | Definition | Reporting options |
|---|---|---|
| Player movement direction: | The movement of the player immediately before they performed the header relative to the ball. | The movement of the player immediately before they performed the header relative to the ball. To be reported using the following options (NB if the player movement direction is coded as jumping, please select one option from both 'Jumping' categories ['Pitch position movement direction' AND 'Body movement direction']. For all other coded player movements select one option from the 'Grounded' category): Grounded: <br> i. No movement: Player who headed the ball was stationary prior to header. This includes upright stance, athletic/squat stance, and a single step used for balance or power generation. <br> ii. Forwards locomotion: Player who headed the ball was engaged in forwards locomotion relative to the direction their body was facing prior to header. <br> iii. Backwards locomotion: Player who headed the ball was engaged in backwards locomotion relative to the direction their body was facing prior to header. <br> iv. Sideways locomotion: Player who headed the ball was engaged in sideways locomotion relative to the direction their body was facing prior to header. <br> v. Turning/pivoting: The player heading the ball was turning/pivoting their body on the spot to face the ball prior to header. <br> vi. Unclassifiable direction: The player heading the ball was moving about a small area in various directions relative to their body prior to header. <br> vii. Unclear: Unable to accurately report. |
| | | Jumping (NB select one option from both pitch position movement direction AND body movement direction): Pitch position movement direction <br> viii. Vertical jump: Player who headed the ball was jumping vertically prior to header, such that their position on the pitch did not change (noticeably) upon landing. <br> ix. Jumping forwards: Player who headed the ball was jumping forwards relative to the direction their body was facing prior to header, such that their starting position was behind them upon landing. <br> x. Jumping backwards: Player who headed the ball was jumping backwards relative to the direction their body was facing prior to header, such that their starting position was in front of them upon landing. <br> xi. Jumping sideways: Player who headed the ball was jumping sideways relative to the direction their body was facing prior to header, such that their starting position was beside them upon landing. <br> xii. Unclear: Unable to accurately report. |
| | | Body movement direction <br> xiii. No body movement: Player who headed the ball remained vertical throughout the jump, such that their head was aligned vertically over their feet. <br> xiv. Forwards body movement: Player who headed the ball leant forwards during the jump, such that their head was aligned anterior to their feet relative to vertical. <br> xv. Backwards body movement: Player who headed the ball leant backwards during the jump, such that their head was aligned posterior to their feet relative to vertical. <br> xvi. Sideways body movement: Player who headed the ball leant sideways during the jump, such that their head was aligned lateral to their feet relative to vertical. <br> xvii. Rotational body movement: Player who headed the ball rotated their body during the jump, such that their body faced a different direction upon landing. <br> xviii. Unclear: Unable to accurately report. |
| Jump height (for contested jumping headers only): | If the header was performed while the player heading the ball was jumping, the height of the jump is recorded in relation to the player closest to them. | To be reported using the following options: <br> i. No jump: The player heading the ball did not jump <br> ii. Jumped higher than other player: The player heading the ball jumped higher than the nearest other player. <br> iii. Did not jump as high as other player: The player heading the ball did not jump as high as the nearest other players who was also jumping. <br> iv. Jumped the same height as other player: The player heading the ball jumped a similar height to the nearest other players who was also jumping. <br> v. Unclear: Unable to accurately report. |

*(Continued)*

| Descriptor | Definition | Reporting options |
|---|---|---|
| Head height: | The height of the player's head when the ball makes contact relative to a normal standing position. | To be reported using the following options:<br>  i. Above chest height: The height of the head when the ball makes contact is above chest height relative to a normal standing position.<br>  ii. Between pelvis and chest height: The height of the head when the ball makes contact is between pelvis and chest height relative to a normal standing position.<br>  iii. At or below pelvis height: The height of the head when the ball makes contact is at or below pelvis height relative to a normal standing position.<br>  iv. Unclear: Unable to accurately report |
| Protective body positioning: | The presence of protective body positioning during a contested header only. | To be reported using the following options (NB more than one option might apply):<br>  i. Protective action – head: The player heading the ball was observed to move their head away from the contest with the likely intent of self-preservation.<br>  ii. Protective action – arms: The player heading the ball was observed to raise/use their arms to protect their space or create distance between them and other players during the contest with the likely intent of self-preservation.<br>  iii. Protective action – turns away: The player heading the ball was observed to turn their body away from the other players in the contest with the likely intent of self-preservation.<br>  iv. No protective actions observed: The player heading the ball was not observed to implement any protective actions with their head/body, positioning, or movement during the contest.<br>  v. Other player pulls out of header: The other player (who was not the player heading the ball) was observed to pull out of their attempt to challenge during the contest when they realised, they were not in a position to win, with the likely intent of self-preservation, and/or preservation of other player/s in the contest.<br>  vi. Unclear: Unable to accurately report |
| Player localisation: | The position of the player heading the ball relative to the nearest other player at the time of the header. | To be reported using the following options:<br>  i. In front of other player/s: The player heading the ball was directly in front of the nearest other player/s.<br>  ii. Behind other player/s: The player heading the ball was directly behind the nearest other player/s.<br>  iii. Next to other player/s: The player heading the ball was directly next to the nearest other player/s.<br>  iv. In between multiple players: The player heading the ball was between multiple other player/s.<br>  v. Diagonally in front of other player/s: The player heading the ball was diagonally in front of the nearest other player/s.<br>  vi. Diagonally behind other player/s: The player heading the ball was diagonally behind the nearest other player/s.<br>  vii. Unclear: Unable to accurately report. |
| Head covering or personal protective equipment: | The player performing the header was observed to be wearing something that covered all or part of their head or face or a form of personal protective equipment. | To be reported using the following options:<br>  i. Yes: specify name of head or face covering (i.e., hijab or face shield)<br>  ii. No: no head or face covering was observed |
| Head movement: | The movement direction of the player's head immediately before they completed the header | To be reported using the following options (NB more than one option might apply):<br>  i. Head facing same direction to movement direction: The player's head was facing the same direction as their movement direction.<br>  ii. Head turned in different direction to movement direction: The player's head was turned in a different direction to their movement direction.<br>  iii. Head facing same direction to body direction: The player's head was facing the same direction as their body, such that their head was centred between their shoulders.<br>  iv. Head turned in different direction to body direction: The player's head was turned in a different direction to their body, such that their head was more rotated towards one shoulder.<br>  v. Unclear: Unable to accurately report. |
| Head impact magnitude: | The peak resultant linear acceleration, rotational acceleration and/or rotational velocity computed from an objective measurement device (such as head impact sensor, instrumented mouthguard or inertial measurement unit) during the header. | To be reported as free text using the data recorded on the measurement device (with the type, make and model of the measurement device as well as data accuracy also being reported). |

*(Continued)*

**Table 3.** (Continued)

| Descriptor | Definition | Reporting options |
|---|---|---|
| Player size: | The size of the player performing the header relative to the other player/s on the pitch. | To be reported using the following options (unless the actual measured player height is available, which should be reported in cm):<br>  i. Same size: The player heading the ball was a similar height to the other player/s<br>  ii. Shorter: The player heading the ball was shorter than other player/s<br>  iii. Taller: The player heading the ball was taller than other player/s<br>  iv. Unclear: Unable to accurately report. |
| Player stability: | How balanced the player heading the ball was at the time of the header. | To be reported using the following options:<br>  i. Player appeared balanced at the time of the header (all non-jumping headers).<br>  ii. Player appeared off-balance at the time of the header (all non-jumping headers).<br>  iii. Player was jumping at the time of the header. The jump (including preparation (take-off), flight, and landing) appeared controlled. |
| Ball tracking: | Whether the player was tracking the in-coming ball immediately prior to the header. | To be reported using the following options:<br>  i. Sufficient ball tracking: The player tracks the ball until head-ball contact. Evident through eyes open and/or head turned in direction of the ball.<br>  ii. Insufficient ball tracking: The player heading the ball makes an attempt to track the ball but stops tracking before expected head-to-ball contact. Evident through eyes closed, and/or head turns away from the ball before head-to-ball contact.<br>  iii. No ball tracking (contested headers only): When the other player in a contested header makes no attempt to track ball movement. Evident by their head being turned away from the ball during the contest.<br>  iv. Unclear: Unable to accurately report. |
| **Match descriptors** | | |
| Match time: | The match time when the header occurred. | To be reported as open text which states the amount of time (in minutes and seconds) from kick-off. Added time should be reported in the first half as 45:00+x:xx and second half as 90:00+x:xx (or equivalent for matches less than 90 minutes). Added time in extra time should be reported as either 115:00+x:xx or 130:00+x:xx) |
| Weather: | The weather at the time of the header. | To be reported using the following options:<br>  i. Dry and sunny: The pitch appears dry, and it is sufficiently sunny to cast shadows on the pitch (*day matches only*)<br>  ii. Dry and cloudy/overcast: The pitch appears dry and there are no obvious shadows on the pitch (in the absence of rain, fog, or snow) or it is a night match.<br>  iii. Wet: Visible rain is observed in the air or there are puddles/wet patches on the pitch *(can be a day or night match)*.<br>  iv. Fog: Poor visibility observed due to weather conditions, in the absence of rain or snow *(can be a day or night match)*.<br>  v. Snow: Visible snow is observed either in the air either or on the pitch *(can be a day or night match)*. |
| Referee: | The action of the referee in response to the heading event. | To be reported using the following options (NB more than one option might apply).<br>  i. Foul for: The referee stops the match and calls a freekick. The foul was committed by the opposition player on the player heading the ball.<br>  ii. Foul against: The referee stops the match and calls a freekick. The foul was committed by the player heading the ball on an opposition player.<br>  iii. Play advantage for: The player heading the ball is fouled by an opposition player. The referee allows for play to continue to the advantage of the team of the player heading the ball.<br>  iv. Play advantage against: The player heading the ball commits a foul on an opposition player. The referee allows for play to continue to the advantage of the opposing team.<br>  v. Yellow card for: The player heading the ball is fouled by an opposition player and the opposition player is given a yellow card.<br>  vi. Yellow card against: The player heading the ball commits a foul on an opposition player and is given a yellow card.<br>  vii. Red card for: The player heading the ball is fouled by an opposition player and the opposition player is given a red card.<br>  viii. Red card against: The player heading the ball commits a foul on an opposition player and is given a yellow card.<br>  ix. VAR stoppage- Play is stopped for VAR review (outcome of VAR review can be recorded using free text)<br>  x. Signal for on-pitch medical assessment: The referee signals to the medical staff to perform an on-pitch assessment on the player heading the ball (or other player/s in a contested header).<br>  xi. End of half/match: Referee blows for the end of the half, match, or other time period (including cooling break).<br>  xii. No referee action<br>  xiii. Unclear: Unable to accurately report. |

*(Continued)*

**Table 3.** (Continued)

| Descriptor | Definition | Reporting options |
|---|---|---|
| Playing surface: | The specific playing surface of the pitch. | To be reported using the following options:<br> i. Grass<br> ii. Artificial turf<br>iii. Hardcourt<br>iv. Gravel<br> v. Sand |

Cognitive Composite [13]. If the data reported from this study [13] are supported and replicated in other cohorts (including women), then it would appear that history of concussion is more deleterious to long-term cognitive function than headers, which has implications for the focus of both head injury prevention initiatives and protection of long-term brain health in football players.

While we recommend that all future heading in football research includes the 27 recommended descriptors, there may be other descriptors from the extended framework that are useful to capture for other purposes. For instance, documenting the time between headers performed by the same player might be considered to inform the design of studies investigating the acute or short-term effects of a single bout of headers on brain health [18]. Specific recommendations on the quality criteria for studies assessing the acute effects of headers have recently been published [6]. Accurately capturing how many headers individual players perform (rather than calculating mean header incidence per player per match based on the numbers of players on the pitch), as well as the time between headers would ensure that studies exploring the short-term effects of headers not only replicate match, or training, conditions, but also consider the ethical implications of requiring players to complete a high number of headers (particularly from high velocity balls) over a short period of time [6].

Additionally, many descriptors recommended for inclusion in minimum reporting criteria overlap with heading performance or technique. For instance, the 3 descriptors: controlled or uncontrolled header, ball-to-head point of contact, and the consequence/outcome of the header. There is limited but emerging research that suggests that head impact magnitude during headers can be reduced by improved technique [14,19]. Therefore, capturing aspects of heading performance or technique alongside incidence data can potentially be used to inform heading coaching frameworks, particularly in young and/or inexperienced players when starting to head the ball. Heading performance can also be used as a metric to distinguish between 'high and low value headers' as a means to reduce heading burden, particularly when focusing on headers more or less likely to lead to a goal, or a turnover of possession [20].

We also recommend that minimum reporting on heading incidence should include raw number of headers per activity (matches, and/or training/ practice sessions) as well as incidence rate per 1000 match (or training) hours, with relevant reference to confidence levels. Where data are collected from matches and training sessions, these data should be reported separately with the exposure hours used to calculate the incidence rate (with 95% confidence intervals) also being reported. While it was outside of the scope of this study to report on the best approach to calculate incidence rates, incidence rates of headers have been presented using match exposure time based on earlier published formulas (number of headers/match exposure time) × 1000), with similar calculations recommended for headers during training [21,22].

## Limitations

While agreed descriptors, definitions, and reporting options have been provided through this project, assessing the feasibility and reliability of recording each of these descriptors was beyond the scope of this study. However, it is recognised that some of the descriptors may be challenging to collect, both when collecting in-person, and particularly when using lower quality footage for review. Furthermore, it is recognised that individual researchers may interpret some options

differently to others. Accordingly, it is expected that intra-rater and/or inter-rater reliability of recording for each included descriptor is completed and reported in any future studies. For minimum reporting descriptors which are not found to have acceptable levels of inter-/intra-rater agreement of >0.80 with an appropriate agreement statistic (e.g., Cohen's Kappa, Fleiss Kappa, Intraclass Correlation Coefficient, percentage agreement) analysts may wish to combine certain reporting options to enhance reliability. For example, ball-to-head point of contact, which can be challenging to distinguish due to factors such as low footage quality and varying hair lines, may have greater reliability by combining the *Forehead* and *Top of head* reporting options.

While the strength of this project was the large number of participants, from a wide geographical area across 4 stakeholder groups, it should be acknowledged that we observed a reduction in completion rate as the project progressed particularly from researchers and players. While the completion rate across all stakeholder groups was > 60% for each of the 5 questionnaires, only 53% completed all 5 questionnaires. Further, it is possible that the strategy used to recruit participants missed some people (particularly players, coaches and medical personnel) with expertise in assessing heading incidence. This may be particularly true for the confederation, CONMEBOL, which had the lowest number of invitees across all 6 confederations. This is important to note for future research to maximise representation across all confederations where this is relevant to the research purpose.

### Considerations for future research

While our Delphi study has provided a comprehensive list of descriptors, definitions and reporting options for future heading in football research, adherence to this framework could be further improved by developing training resources using match video examples to highlight how this coding should be applied, which in turn should improve intra- and inter-rater reliability of reporting, further improving research quality. While our descriptors did include whether a header resulted in a potential or diagnosable head injury, further details regarding potential injury events are not fully captured with the included descriptors, particularly if the injury related to a non-header related head impact. For these events, it is suggested that researchers consider the FIFA Football Language with a Medical Extension for coding injury inciting circumstances [23–25]. Finally, as heading technique and performance is an emerging area of scientific enquiry, further exploring additional descriptors related to these aspects of heading might be a useful addition to the literature.

### Conclusion

This Delphi project gained consensus from participants across a range of relevant football backgrounds including coaches, medical personnel, players, and researchers, on a list of 27 descriptors, their definitions, and reporting options, that should be included in minimum reporting criteria for heading in football research. A further list of 27 descriptors also achieved agreed definitions, and reporting options. Importantly, a standardised operational definition of a header was agreed, with recommendations that minimum reporting on heading in football research should include raw numbers of headers per activity (matches, and/or training/ practice sessions) as well as incidence rate per 1000 match (or training) hours. Collectively, these recommendations should improve the quality in reporting of heading in football data and enable comparison of header exposure data between studies regardless of when or where the data were collected.

> Key points:
>
> - This Delphi study aimed to establish a consensus on descriptors, definitions, and reporting methods for heading in football research to improve consistency and quality when exploring the incidence and characteristics of headers.
>
> - We involved 167 participants from diverse football-related backgrounds, including coaches, players, medical personnel, and researchers.

- Consensus was achieved on 27 descriptors for minimum reporting criteria and an additional 27 descriptors for an expanded framework. These descriptors cover various aspects such as player demographics, type of header, head injury, and non-header related head impacts.

- We defined a header as "a head-to-ball contact where the player makes a deliberate movement to redirect the trajectory of the ball using their head." This definition helps differentiate between intentional headers and unintentional ball-to-head impacts.

- We recommend that future heading in football research includes raw number of headers per activity as well as incidence rate per 1000 match (or training) hours. Where data are collected from both matches and training sessions, these data should be reported separately with the exposure hours used to calculate the incidence rate also being reported.

- Furthermore, researchers should document the reliability (intra- and/or inter-rater) for each descriptor reported.

## Supporting information

**S1 Appendix. Additional project data.**
(DOCX)

## Author contributions

**Conceptualization:** Kerry Peek, Andreas Serner.

**Data curation:** Kerry Peek, Andrew G Ross.

**Formal analysis:** Kerry Peek.

**Funding acquisition:** Kerry Peek.

**Investigation:** Kerry Peek, Andrew G Ross, Julia Georgieva, Thor Einar Andersen, Tim Meyer, Vincent Gouttebarge, Sara Dahlen, Andreas Serner.

**Methodology:** Kerry Peek, Andrew G Ross, Paula R Williamson, Julia Georgieva, Thor Einar Andersen, Tim Meyer, Vincent Gouttebarge, Sara Dahlen, Mike Clarke, Andreas Serner.

**Project administration:** Kerry Peek.

**Validation:** Kerry Peek, Andrew G Ross.

**Writing – original draft:** Kerry Peek.

**Writing – review & editing:** Andrew G Ross, Paula R Williamson, Julia Georgieva, Thor Einar Andersen, Tim Meyer, Vincent Gouttebarge, Sara Dahlen, Mike Clarke, Andreas Serner.

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
