## [Decision Letter · Decision Letter 0]

Dear Dr. Peek,

Thank you for submitting your manuscript to PLOS ONE. After careful consideration, we feel that it has merit but does not fully meet PLOS ONE’s publication criteria as it currently stands. Therefore, we invite you to submit a revised version of the manuscript that addresses the points raised during the review process.

We look forward to receiving your revised manuscript.

Kind regards,

Mario Lopes, Ph.D

Academic Editor

PLOS ONE

**Journal Requirements:**

1. When submitting your revision, we need you to address these additional requirements. Please ensure that your manuscript meets PLOS ONE's style requirements, including those for file naming. The PLOS ONE style templates can be found at https://journals.plos.org/plosone/s/file?id=wjVg/PLOSOne_formatting_sample_main_body.pdf and https://journals.plos.org/plosone/s/file?id=ba62/PLOSOne_formatting_sample_title_authors_affiliations.pdf 2. Thank you for stating in your Funding Statement: This project was part funded by a Thompson Equity Prize (The University of Sydney) awarded to KP. Please provide an amended statement that declares *all* the funding or sources of support (whether external or internal to your organization) received during this study, as detailed online in our guide for authors at http://journals.plos.org/plosone/s/submit-now.  Please also include the statement “There was no additional external funding received for this study.” in your updated Funding Statement. Please include your amended Funding Statement within your cover letter. We will change the online submission form on your behalf. 3. Thank you for stating the following in the Competing Interests section: AS and KP declare employment with FIFA. KP and JG have been contracted Injury Spotters for FIFA organised tournaments (2023). KP, TM, VG and AS are members of FIFA’s Heading Expert Group. TM is chairman of UEFA´s and the German FA´s (DFB) Medical Committee as well as of the German League Organisation´s (DFL) working group “Medicine in Professional Football”. TEA is Chief Medical Officer of the Football Association of Norway (NFF). VG is the Chief Medical Officer at FIFPRO. All authors declare no other relevant financial or non-financial competing interests.   We note that one or more of the authors are employed by a commercial company.  a. Please provide an amended Funding Statement declaring this commercial affiliation, as well as a statement regarding the Role of Funders in your study. If the funding organization did not play a role in the study design, data collection and analysis, decision to publish, or preparation of the manuscript and only provided financial support in the form of authors' salaries and/or research materials, please review your statements relating to the author contributions, and ensure you have specifically and accurately indicated the role(s) that these authors had in your study. You can update author roles in the Author Contributions section of the online submission form. Please also include the following statement within your amended Funding Statement. “The funder provided support in the form of salaries for authors, but did not have any additional role in the study design, data collection and analysis, decision to publish, or preparation of the manuscript. The specific roles of these authors are articulated in the ‘author contributions’ section.”If your commercial affiliation did play a role in your study, please state and explain this role within your updated Funding Statement.  b. Please also provide an updated Competing Interests Statement declaring this commercial affiliation along with any other relevant declarations relating to employment, consultancy, patents, products in development, or marketed products, etc.   Within your Competing Interests Statement, please confirm that this commercial affiliation does not alter your adherence to all PLOS ONE policies on sharing data and materials by including the following statement: "This does not alter our adherence to  PLOS ONE policies on sharing data and materials.” (as detailed online in our guide for authors http://journals.plos.org/plosone/s/competing-interests) . If this adherence statement is not accurate and  there are restrictions on sharing of data and/or materials, please state these. Please note that we cannot proceed with consideration of your article until this information has been declared. Please include both an updated Funding Statement and Competing Interests Statement in your cover letter. We will change the online submission form on your behalf. 4. We note that this data set consists of interview transcripts. Can you please confirm that all participants gave consent for interview transcript to be published? If they DID provide consent for these transcripts to be published, please also confirm that the transcripts do not contain any potentially identifying information (or let us know if the participants consented to having their personal details published and made publicly available). We consider the following details to be identifying information:- Names, nicknames, and initials- Age more specific than round numbers- GPS coordinates, physical addresses, IP addresses, email addresses- Information in small sample sizes (e.g. 40 students from X class in X year at X university)- Specific dates (e.g. visit dates, interview dates)- ID numbers Or, if the participants DID NOT provide consent for these transcripts to be published:- Provide a de-identified version of the data or excerpts of interview responses- Provide information regarding how these transcripts can be accessed by researchers who meet the criteria for access to confidential data, including:a) the grounds for restrictionb) the name of the ethics committee, Institutional Review Board, or third-party organization that is imposing sharing restrictions on the datac) a non-author, institutional point of contact that is able to field data access queries, in the interest of maintaining long-term data accessibility.d) Any relevant data set names, URLs, DOIs, etc. that an independent researcher would need in order to request your minimal data set. For further information on sharing data that contains sensitive participant information, please see: https://journals.plos.org/plosone/s/data-availability#loc-human-research-participant-data-and-other-sensitive-data If there are ethical, legal, or third-party restrictions upon your dataset, you must provide all of the following details (https://journals.plos.org/plosone/s/data-availability#loc-acceptable-data-access-restrictions):a) A complete description of the datasetb) The nature of the restrictions upon the data (ethical, legal, or owned by a third party) and the reasoning behind themc) The full name of the body imposing the restrictions upon your dataset (ethics committee, institution, data access committee, etc)d) If the data are owned by a third party, confirmation of whether the authors received any special privileges in accessing the data that other researchers would not havee) Direct, non-author contact information (preferably email) for the body imposing the restrictions upon the data, to which data access requests can be sent 5. When completing the data availability statement of the submission form, you indicated that you will make your data available on acceptance. We strongly recommend all authors decide on a data sharing plan before acceptance, as the process can be lengthy and hold up publication timelines. Please note that, though access restrictions are acceptable now, your entire data will need to be made freely accessible if your manuscript is accepted for publication. This policy applies to all data except where public deposition would breach compliance with the protocol approved by your research ethics board. If you are unable to adhere to our open data policy, please kindly revise your statement to explain your reasoning and we will seek the editor's input on an exemption. Please be assured that, once you have provided your new statement, the assessment of your exemption will not hold up the peer review process. 6. Your ethics statement should only appear in the Methods section of your manuscript. If your ethics statement is written in any section besides the Methods, please delete it from any other section.

**Additional Editor Comments:**

Dear Authors,

The reviewers have made interesting comments that will help enhance your manuscript. Please follow closely their recommendations.

Reviewers' comments:

Reviewer's Responses to Questions

**Comments to the Author**

1. Is the manuscript technically sound, and do the data support the conclusions?

Reviewer #1: Yes

Reviewer #2: Yes

2. Has the statistical analysis been performed appropriately and rigorously?

Reviewer #1: N/A

Reviewer #2: N/A

3. Have the authors made all data underlying the findings in their manuscript fully available?

Reviewer #1: Yes

Reviewer #2: Yes

4. Is the manuscript presented in an intelligible fashion and written in standard English?

Reviewer #1: Yes

Reviewer #2: Yes

**Reviewer #1: ** This original research paper describes a Delphi study to develop a method of develop a definition of ‘heading of the ball’ (i.e., heading) in soccer and a protocol to assess the skill characteristics for the purposes of injury surveillance and technique analysis. The protocol is designed to be applied primarily to video, but other data sources (e.g., accelerometery) are also included. As far as I can determine, the study has not been published elsewhere.

The participants and methods used for the Delphi study are appropriate, largely concordant with published guidelines (ACCORD and CREDES) and reported in sufficient detail. One hundred and sixty-seven 167 participants with experience in research and/or soccer who commenced the Delphi study. The Delphi study consisted of five rounds over three phases to select the characteristics that should comprise the minimum analysis, and determine definitions. The first phase determined the fields (described as descriptors) for the analysis, the second phase defined the fields while the final phase determined the method or list of field variables used to assess/describe each field. Of the five rounds of consensus, 50% of participants completed all 5 rounds. Although the attrition is of some concern, 80+ participants is still substantial and efforts are made to report on the characteristics of respondents at each step. Subsequently, 27 fields were selected for inclusion in the minimal analysis and a further 27 characteristics, deemed not as important by participants in the Delphi study, were also defined for an expanded analysis.

The described protocol will provide a standardised approach to identifying a header and investigating characteristics of heading in soccer from video, allowing the incidence of heading and the characteristics analysed to be determined and, when combined with injury surveillance data, potentially allowing for causal inference. Whilst there are some characteristics which I expect will be challenging to evaluate reliably from video, this is beyond the scope of the current study and the authors have wisely encouraged reporting of reliability statistics. It is my opinion that the manuscript meets the aims and scope of PLoSOne and would be of interest to its readers, subject to consideration of several revisions.

Firstly, I would encourage the authors to thoroughly review and edit the manuscript to improve its readability. There were many instances where the language used is unclear. I provide three of the many examples throughout the manuscript:

“Heading incidence research could be explored as one approach of assessing the potential causal inference between heading and long-term issues with brain health” - Does you mean that you wish to explore heading incidence research or that you want to investigate the incidence of heading and heading characteristics? I believe it's the latter but the expression is overly complex. Similarly, "our understanding of heading risk then THIS needs" (lines122-3). What do you mean by THIS?, please be specific so that it is unambiguous for the reader. Finally "Research methods experts, having... consensus projects" (lines 139-40) could be simplified as "Delphi and consensus study experts", or similar. Please be clear and explicit throughout. Additionally, there are many typos throughout which need correction (e.g., “insures” vs ensures (line 116) or “can” vs could (line 118), “whereby” vs thereby (line 441), the repeated phrase on line 468, etc.).

Secondly, the introduction should be expanded to better position this work with respect to previous literature. The authors could better introduce the discussion in the field regarding ball heading and its long-term health impacts and define also define the difference between heading and non-heading impacts. From the context and literature, I suspected non-heading impacts may refer to be head-to-head contact and it wasn’t clear until very late in the manuscript that this wasn’t the case. The introduction could also briefly introduce efforts from soccer and other sports to define injury events and develop qualitative analysis methods (e.g., Tenga’s work, volleyball, tackling in rugby union etc.), what was good and what could be improved. This would then allow a discussion point on how this study improved on earlier studies, thus broadening the relevance of this study beyond heading incidence.

Thirdly, relating to clarity of reporting, I note that the project was part funded through an academic prize at an Australian university although ethics approval was obtained in Switzerland. Were there other sources of funding (e.g., FIFA) which should be included?

The direction to include reliability in studies reporting incidence with the protocol is a good one. There are several instances where Cohen’s Kappa is not appropriate, such as continuous data, multiple raters (i.e., > 2) or where one variable occurs at a substantially higher frequency than others. For this reason, the authors might consider something like “agreement of >0.80 with an appropriate agreement statistic (e.g., cohen’s Kappa, Fleiss Kappa, ICC, percentage agreement, etc.)”

Finally, the reason for hijab being bracketed with protective equipment seems unusual and needs to be better explained to the reader. The other items listed on page 16 seem have an obvious relationship with skill and performance and so need little justification. Head covering on the other hand states that participants suggested that it be included but why it is important wasn’t mentioned in this article or in the associated systematic review. Did the expert panel form the view that a hijab attenuated head force and reduce the risk of injury or might obscure vision and increase the risk of injury etc?

**Reviewer #2: ** Firstly, I would like to congratulation the authors for undertaking a comprehensive Delphi study. A strength of this study is the inclusion of an large number of participants.

My major comment/concern is regarding the focus on ‘incidence’ – but then limited explanation of how one should accurately and robustly collect exposure. Given exposure time is the denominator for incidence, this will have a large impact. Especially in training – exposure could be continuous (start – end), drill duration (removal of ‘dead time’) or player involvement in the drills. I think it is the authors either point the reader to where this exists, and recommend the use of an existing definition, or include this.

I do wonder if this is more than incidence – for examples some of definitions are beyond heading incidence. Isn’t heading incidence only one part of the story, especially when considering the results of this study even say that magnitude is important. Therefore, isn’t this a consensus for heading studies?

I was also surprised that a head-head collision (during a header) wasn’t included. I’m not sure if this was deliberately excluded (based on the study design), or no one thought this was important, but it does feel like it warrants more discussion. E.g., if we know the head-head collision rates during different heading activities / in different cohorts – that is very helpful when considering ‘short-term’ head injury risk? This was discussed briefly in lines 476-478, and 540-543, but it seems an omission.

My minor comments are;

At first mention (in abstract and in main documents), report ‘football (i.e., soccer)’

You report short and long-term risk in abstract, but don’t define them in the introduction. I would suggest reporting the ‘short-, medium-, and longer-term risk’…. (and defining).

Key points – include clarify what is the denominator for calculating 1000 hours.

Key points (line 108). ‘Furthermore’, not ‘further’.

Introduction (line 119). I’m not sure what is encouraging? I couldn’t follow what ‘This is’ was referring to.

Introduction (line 124). A recent systematic review needs a reference

Method – I didn’t find the method very clear early on, and think a study design or overview would be helpful. For example, the ‘each objective and phase (1-3) on line 176 came out of the blue, and I didn’t know what each objective was, as the objective (singular – not pleural) on line 129-131 is set out as one objective. Figure 1 needs to do a better job of clarifying what the three phases are – which are nicely summarised on line 177-179.

Method (line 153). In the journal information, you report University of Sydney as the ethics organisation, then Swiss Association…. In the main text. Please clarify.

Results (line 315) The abbreviations (e.g., CAF, CONCACAF) on line 315 come before they are written in full (lines 330-333).

Results (lines 326-328) The percentages reported add up to 99%

Table 1. The numbers (112, 67%) in the last row of the men or women column, don’t follow what were previous reported, therefore aren’t intuitive. It would be useful to report men and women?

Limitations (line 474, 516, 529). ‘Furthermore’, not ‘further’.

Discussion (line 507) expand on how to capture exposure as the denominator.

**Do you want your identity to be public for this peer review?** For information about this choice, including consent withdrawal, please see our Privacy Policy

Reviewer #1: No

Reviewer #2: **Yes: ** Ben Jones

---

## [Author Response · Author response to Decision Letter 1]

10 Jun 2025

We have included our response to reviewer comments as a separate file.

---

## [Editor Report · Decision Letter 1]

Consensus on the descriptors, definitions, and reporting methods for heading in football studies:  A Delphi study

PONE-D-25-11018R1

Dear Dr. Peek,

We’re pleased to inform you that your manuscript has been judged scientifically suitable for publication and will be formally accepted for publication once it meets all outstanding technical requirements.

Kind regards,

Mario Lopes, Ph.D

Academic Editor

PLOS ONE

Additional Editor Comments (optional):

Dear Authors,

The authors have attended to all suggestions made by the reviewers. I congratulate the effort placed to enhance this manuscript. I consider the manuscript ready for publishing.
---

## [Editor Report · Acceptance letter]

PONE-D-25-11018R1

PLOS ONE

Dear Dr. Peek,

I'm pleased to inform you that your manuscript has been deemed suitable for publication in PLOS ONE. Congratulations! Your manuscript is now being handed over to our production team.

Kind regards,

on behalf of

Prof. Mario Lopes

Academic Editor

PLOS ONE